# Treatment of Pelvic Organ Prolapse by the Downregulation of the Expression of Mitofusin 2 in Uterosacral Ligament Tissue via Mesenchymal Stem Cells

**DOI:** 10.3390/genes13050829

**Published:** 2022-05-06

**Authors:** Xiaoqing Wang, Ruiju He, Songwen Nian, Bingbing Xiao, Yu Wang, Lei Zhang, Xiaoxiao Wang, Ruilin Guo, Ye Lu

**Affiliations:** Department of Obstetrics and Gynecology, Peking University First Hospital, Beijing 100034, China; wxq0722@bjmu.edu.cn (X.W.); rae172025@163.com (R.H.); 1911210236@bjmu.edu.cn (S.N.); 216108558@bjmu.edu.cn (B.X.); moon_snowrabbit@163.com (Y.W.); zhanglei801440@163.com (L.Z.); 1611210275@bjmu.edu.cn (X.W.); 2011210202@bjmu.edu.cn (R.G.)

**Keywords:** pelvic organ prolapse, bone marrow stem cells, mitofusin 2, procollagen, uterosacral ligament, rat model

## Abstract

Background: The relationship between pelvic organ prolapse (POP), an aging-related disease, and the senescence-related protein mitofusin 2 (Mfn2) has rarely been studied. The aim of the present study was to explore the therapeutic effects of the downregulation of Mfn2 expression by stem cells on POP through animal experiments. Methods: First, a rat POP model was constructed by ovariectomy and traction. The rats in the non-pelvic organ prolapse (NPOP) and POP groups were divided into four groups for negative controls (N1–N4, N1: NPOP-normal saline; N2: NPOP-untransfected stem cells; N3: NPOP-short hairpin negative control (NPOP-sh-NC); N4: NPOP-short hairpin-Mfn2 (NPOP-sh-Mfn2)), and four groups for prolapse (P1–P4, P1: POP-normal saline; P2: POP-untransfected stem cells; P3: POP-sh-NC; P4: POP-sh-Mfn2), respectively. Stem cells were then cultured and isolated. The expression of Mfn2 was inhibited by lentivirus transfection, and the stem cells were injected into the uterosacral ligament of the rats in each group. The expression levels of Mfn2 and procollagen 1A1/1A2/3A1 in the uterosacral ligaments of the rats were observed at 0, 7, 14, and 21 days after injection. Results: Compared to the rats in the NPOP group, the POP rats had significant prolapse. The Mfn2 expression in the uterosacral ligaments of the POP rats was significantly increased (*p* < 0.05, all), and the expression of procollagen 1A1/1A2/3A1 was significantly decreased (*p* < 0.001, all). The POP rat model maintained the same trend after 21 days (without stem cell injection). At day 14, compared to the rats in the N1 group, the Mfn2 expression in the uterosacral ligament of the rats in the N4 group was significantly decreased (*p* < 0.05, all), and the expression of procollagens was significantly increased (*p* < 0.05, all). Similarly, compared to the rats in the P1 group, the Mfn2 expression in the uterosacral ligament of the rats in the P4 group was significantly decreased (*p* < 0.05, all), and the expression of procollagens was significantly increased (*p* < 0.05, all). Similarly, on day 21, the Mfn2 mRNA and protein expression in the uterosacral ligament of the POP and NPOP rats was significantly decreased (*p* < 0.05, all), and the expression of procollagens was significantly increased (*p* < 0.05, all) in the rats in the sh-Mfn2 group (N4, P4) compared to the rats in the saline group (N1, P1). Conclusions: The downregulation of Mfn2 expression by stem cells decreased the expression of Mfn2 and increased the expression of procollagen1A1/1A2/3A1 in the uterosacral ligament of the POP rats; this effect was significant 14–21 days after the injection. Thus, Mfn2 may be a new target for POP control.

## 1. Introduction

The incidence of pelvic organ prolapse (POP), an age-related disease, is high and shows an increasing annual trend [1,2]. POP affects people’s quality of life because of its high operation cost [3], many postoperative complications [3,4], symptoms affecting the quality of life [5,6], unsatisfactory conservative treatment effect [7,8,9], and so on. Among the existing studies, there are few basic studies related to the pathological process of POP, such as cell behavior [10], histological characteristics [11], protein expression changes [12,13] and cell therapy [14]. Mitofusin 2 (Mfn2), also known as the hyperplasia suppression gene (HSG), mediates mitochondrial fusion and signal transduction, and its abnormal expression affects the process of aging [15,16]. At the current time, studies on Mfn2 are mostly focused on cardiovascular and cerebrovascular diseases, diabetes, and cancer, and there is a lack of research on the relationship between Mfn2 and POP. As both Mfn2 and POP are related to aging, the changes in Mfn2 in fibroblasts from pelvic floor tissues were examined in our previous studies, and the correlation between the two was confirmed through in vivo and cell experiments [17,18]. We found that Mfn2 was abnormally elevated in uterosacral ligament tissues and fibroblasts in POP patients, and the expression of procollagen 1A1/1A2/3A1 was significantly reduced. Our previous studies [19,20] also confirmed that, after the downregulation of Mfn2 expression, the expression of procollagen 1A1/1A2/3A1 increased and the proliferation of fibroblasts increased, which may contribute to the strengthening of the pelvic floor support structure dominated by the uterosacral ligament. Therefore, we speculated that the downregulation of Mfn2 expression could treat POP [20]. However, this finding lacks evidence from animal studies.

The existing treatment methods for POP are surgery, pessary, and pelvic floor rehabilitation. However, surgical treatment has a high recurrence rate [3,4] and multiple complications of the mesh [5]. Pessaries can easily cause infections, vaginal wall erosion, increased vaginal secretions, and other problems [7], and pelvic floor rehabilitation training has a poor efficacy [8]. Therefore, it is important to explore new, safer, and more effective treatment methods. Bone marrow mesenchymal stem cells (BMSCs) have attracted much attention owing to their ability of self-replication and directed differentiation [21,22]. Stem cells, such as induced pluripotent stem cells [23,24], reprogrammed fibroblasts [25], endometrial stem progenitor cells [26], and regenerative medicine offer many possibilities for the treatment of clinical diseases. However, to date, the clinical application of stem cells has been limited to a few diseases [23]. Compared with traditional gene regulation methods, stem cells are safer and more conducive to clinical transformation. There have been preliminary explorations of the treatment of POP using stem cells [14,25], but these studies focused more on the combination of stem cells and biodegradable scaffolds, and there are few studies on the targeted regulation of pathogenic genes at the stem cell level, especially regarding the treatment of POP after the regulation of stem cell Mfn2 expression.

Here, we aimed to evaluate the effect of interfering with Mfn2 expression at the stem cell level in a rat POP model. The effects of stem cells and their associated gene therapy on POP can be evaluated and the therapeutic value of Mfn2 expression can be assessed.

## 2. Materials and Methods

### 2.1. Construction of the POP Animal Model

Female SPF rats (8 weeks old) were purchased from Beijing Vital River Laboratory Animal Technology Co., Ltd. (Beijing, China). The rats were randomly divided into the POP (*n* = 48) and non-pelvic organ prolapse (NPOP, *n* = 48) groups after entering the animal center. As POP is a disease associated with middle and old age, all of the rats were castrated to mimic the hormone levels in POP patients [14]. In brief, after anesthesia with intraperitoneally administered ketamine (120 mg/kg), a longitudinal incision was made 0.5–1 cm from both sides the midline of the back, located roughly 0.5–1 cm downward at the angle between the costal arch and the spinal edge of the rat. After separating the skin, fascia, and muscles, the deep-red fallopian tube was located and followed to the ovary. Toothless forceps were inserted into the abdominal cavity, and the ovary and the end of the fallopian tube were gently removed from the abdominal cavity [14]. Small hemostatic forceps were clamped at the smallest location between the end of the uterus and the ovary. After the ligation of “3-0” operation suture, the ovaries were completely removed using ophthalmic scissors. After releasing the hemostatic forceps, the uterus was gently returned to the abdominal cavity, and penicillin powder was placed in the abdominal cavity. The muscle and skin were sutured, and the incision was cleaned with alcohol. The POP rat model was established as follows: after anesthesia, the rats were secured on the operating table. An 18F urethral catheter was inserted into the vagina of the rats and fixed into the cervix with a 3-0 suture. Saline (2.5–3 mL) was injected into the bladder of the catheter. The lower part of the catheter was connected to a 100 g weight to pull the uterus 4 h a day, which was performed every 3 days. This was repeated ten times in order to establish the POP model. The animals were sacrificed by excess CO_2_ inhalation. The animal experiments were approved by the Welfare Ethics Review of Animal Experiments at Peking University First Hospital (acceptance number: 201838), and the operator’s certificate for animal experiments was 1116112800082.

### 2.2. Stem Cell Culture

Rat BMSCs were purchased from ScienCell, Inc. (catalog #R7500, San Diego, CA, USA). In brief, T-75 culture bottles (Corning, Corning, NY, USA) were prepared overnight at 37 °C and 5% CO_2_ with 5 mL PBS + 150 μL fibrin lectin solution (catalog #8248)) before use. The solution was then aspirated and added to the complete medium (mesenchymal stem cell medium, catalog #7501, Sciencell, San Diego, CA, USA) to a volume of 15 mL in a sterile tray. The frozen cells were placed in a water bath at 37 °C until they were completely thawed. The cells were transferred into culture flasks at a density of 5,000 cells/cm^2^, and were observed under a microscope (OLYMPUS, Tokyo, Japan). The medium was changed every 24–48 h. When the cells reached 90% confluency, they were trypsinized with 0.25% trypsin (T/E, catalog #0103, Sciencell, San Diego, CA, USA) for 1–2 min until the cells were completely round. A medium was added, and the cells were centrifuged at 1000 rpm for 5 min, before being resuspended in medium and transferred to a fulfillment-enveloped culture flask. The cells were harvested for further analysis once they reached 80% confluency.

### 2.3. Identification of the Stem Cells

When the stem cells were passaged to the fifth generation, the cell phenotype was identified by flow cytometry. After trypsinization, the cells were centrifuged at 1000 rpm for 10 min and washed twice with PBS. After resuspension in PBS, the cells were centrifuged at 2000 rpm for 6 min, and the supernatant was discarded. One microliter of antibody was added to each tube, and was incubated for 30 min at 4 °C in the dark. Then, 1 mL PBS was added to each tube and centrifuged at 2000 rpm for 6 min. This step was repeated twice. Finally, the cells were resuspended in 0.5 mL PBS and analyzed using flow cytometry. The antibodies included positive markers (CD29 (catalog #102207, Biolegend, San Diego, CA, USA), CD90 (catalog #202503, Biolegend, San Diego, CA, USA), CD106 (catalog #200403, Biolegend, USA), and negative markers, CD11b (catalog #201807, Biolegend, San Diego, CA, USA), and CD45 (catalog #202207, Biolegend, San Diego, CA, USA). According to FlowJo software (7.6.5, Ashland, Wilmington, DE, USA) analysis, CD29+, CD90+, and CD106+ cells were designated as MSCs, and CD45+ and CD11b+ cells were designated as hematopoietic stem cells.

### 2.4. Differentiative Capacity of MSCs

In order to identify the lipid-forming capacity of the MSCs, we used the dye Oil Red O (Sciencell, San Diego, CA, USA. Oil Red Straining Kit No.0843). MSCs were seeded into 6-well plates at a density of 105 cells/well. After the growth and fusion of the MSCs, adipogenic medium (50 mL Adipogenic MSCM, 5 mL FBS, 0.5 mL double antigens, and 0.5 mL MSCGs; Sciencell, 7541, San Diego, CA, USA, Mesenchymal Stem Cell Adipogenic Differentiation Medium) was added. The control group was cultured in normal medium (mesenchymal stem cell medium, catalog #7501, Sciencell, San Diego, CA, USA). Oil Red O staining was performed 2 weeks after the induction of adipogenesis. After the cells were washed with PBS, they were fixed with a kit fixative for 15 min. The fixative solution was removed by washing with DiH_2_O three times. After the addition of the Oil Red O dye, the solution was incubated at room temperature for 15 min, and subsequently washed with DiH_2_O five times. The cells were observed by microscopy, and five images were randomly selected from each sample.

Alizarin Red S (Sciencell, Alizarin Red S Staining Kit, No.0223, Sciencell, San Diego, CA, USA) staining was used to identify the osteogenic ability of the MSCs. We inoculated MSCs in 6-well plates at a density of 10^5^ cells/well. After growth and integration, the experimental group was treated with MSC Osteogenic Differentiation Medium (MODM, catalog #7531, San Diego, CA, USA), while the control group was treated with conventional medium (mesenchymal stem cell medium, catalog #7501, Sciencell, San Diego, CA, USA). After 3 weeks of osteogenic induction, Alizarin Red S staining was performed. The cells were washed with PBS and fixed with 4% paraformaldehyde for 15 min. The fixative was removed by washing with DiH_2_O three times. Next, 1 mL Alizarin Red S dye solution was added to each well and incubated at room temperature for 20–30 min, then washed with DiH_2_O five times. The cells were observed by microscopy, and five images were randomly selected from each sample.

### 2.5. Construction and Transfection of the Lentiviral Vector

The experimental lentivirus with GFP tag was designed and constructed by Genechem (Shanghai Genechem Co., Ltd., Shanghai, China). Interfering shRNA sequences were designed and synthesized according to the ORF sequence of the Mfn2 gene, which was packaged within the lentivirus and transfected into BMSCs, and the cells were divided into two groups: the short hairpin-negative control (sh-NC) and the short hairpin-Mfn2 (sh-Mfn2). The lentiviruses and BMSCs were co-cultured. Before transfection, different concentrations of puromycin were used for screening, and the appropriate concentration of puromycin was determined to be 5 μg/mL (all of the stem cells were killed within 3 days). The GFP expression in the cells was observed under a fluorescence microscope 96 h after viral infection, and the cells were retained for subsequent detection. The purified cells were transformed and passaged in a medium containing 5 μg/mL puromycin.

### 2.6. The Injection of the Stem Cells

The NPOP and POP rats were divided into eight groups, correspondingly: N1: NPOP-normal saline; N2: NPOP, untransfected stem cells; N3: NPOP-sh-NC; N4: NPOP-sh-Mfn2; P1: POP-normal saline; P2: POP, untransfected stem cells; P3: POP-sh-NC; P4: POP-sh-Mfn2. Stem cells from each group were cultured to 70–80% confluency in a sterile environment, trypsinized, and counted with 0.25% trypsin. During the injection treatment, stem cells (10^6^ cells/0.5 mL/mouse) or an equivalent amount of standard saline were injected into the uterosacral ligament of the rats through the vagina.

### 2.7. Real-Time Quantitative PCR (qRT-PCR)

In order to detect the mRNA expression of Mfn2 and procollagen in the cells and tissues, qRT-PCR was performed. The total RNA from the cells or the uterosacral ligament was lysed on ice, using TRIzol. Next, we extracted RNA with organic solvents such as chloroform, then precipitated, washed, air dried, and finally dissolved the RNA. A microplate reader (Bio Tek, Winooski, VT, USA) was used to measure the concentration of RNA. Then, the total RNA was reverse transcribed into cDNA using a Transcript One-Step gDNA Removal and cDNA Synthesis Supermix (AT311-03, TransGen Biotech, Beijing, China) kit. The amplification of Mfn2, procollagen 1A1/1A2/3A1, and β-actin was performed by qRT-PCR using the ABI Power SYBR Green Gene Expression System (Applied Biosystems, Waltham, MA, USA). The relative expression of target genes was calculated using the 2^−ΔΔCT^ method. The primer sequences were as follows: β-actin forward 5′-CCGCGAGTACAACCTTCTTG-3′, reverse 5′-CGTCATCCATGGCGAACTGG-3′; Mfn2: forward: 5′-AGCAAGACATGATAGACGGCTT-3′; reverse 5′-AGACAGTGGGTGCTTTCCTTC-3′; procollagen1A1: forward: 5′-TCTGACTGGAAGAGCGGAGA-3′; reverse: 5′-GGTGGGAGGGAACCAGATTG-3′; procollagen1A2: forward: 5′-TGTCGATGGCTGCTCCAAAA-3′; reverse: 5′-CCGATGTCCAGAGGTGCAAT-3′; procollagen3A1: forward: 5′-TGGGAAAGGTGAAATGGGTCC-3′; reverse: 5′- CTTTGCTCCATTCTTGCCCG-3′.

### 2.8. Western Blot

Western blot analysis was used to detect the protein expression levels in the cells and uterosacral ligaments. Briefly, the total cell protein was extracted using RIPA (Radio Immunoprecipitation Assay), the protein concentration was measured by the BCA (bicinchoninic acid) method, and 20 μg protein was subjected to gel electrophoresis. Following protein transfer, the membranes were incubated with anti-Mfn2 (1:500, catalog #AB56889, Abcam, Cambridge, UK), anti-procollagen 1A1/1A2/3A1 (anti-procollagen1A1: 1:1000, catalog #OM164818, OmnimAbs, Alhambra, CA, USA; anti-procollagen1A2: 1:2000, catalog #SC-166572, Santa Cruz, CA, USA; anti-procollagen3A1: 1:2000, catalog #SC-166333, Santa Cruz, CA, USA), and anti-β-actin (1:2000, catalog #TA-09, ZSGB-Bio, Beijing, China). The next day, after three washes with TBST, horseradish peroxidase (HRP)-labeled goat anti-mouse IgG (1:5000; catalog #ZB-2305, ZSGB-Bio, Beijing, China), and HRP-labeled goat anti-rabbit IgG (H + L) (1:5000; catalog #ZB-2301, ZSGB-Bio, Beijing, China) were incubated at room temperature for 1 h. Immunoblotting chemiluminescence imaging was performed using an HRP chemical substrate (Immobilon Western Chemilum HRP Substrate, WBKLS0100, Millipore, Darmstadt, Germany). ImageJ software (1.8.0., NIH, Bethesda, MD, USA) was used to calculate the grayscale ratio of the target protein bands to the β-actin bands.

### 2.9. Statistical Analysis

All of the data were analyzed using SPSS23 software (IBM, Armonk, NY, USA). One-way analysis of variance (ANOVA) and independent sample t-tests were used for the intra-group comparison of the data conforming to normality. For data that did not conform to normality, the Kruskal–Wallis test was used for comparison between the groups. Statistical significance was set at *p* < 0.05.

## 3. Results

### 3.1. Rat Model of POP

The construction of the POP rat model (*n* = 48) formed the basis of the animal experiments. In this study, compared with the one and five times catheter traction and dilation group, the POP rat model was successfully established and showed the characteristics of prolapse by castration surgery combined with the 10 times catheter traction and dilation group (Figure 1A). After 10 trials, the rats showed significant prolapse symptoms. Then, the rats in the POP and NPOP groups were sacrificed, and uterosacral ligaments were retained for qRT-PCR and Western blot detection. Compared to the rats in the NPOP group, we found (Figure 1B) that on day 0, the mRNA level of the Mfn2 in the rats in the POP group was significantly increased (*n* = 3, *p* < 0.001), and the expression of procollagen 1A1/1A2/3A1 was significantly decreased (procollagen1A1: *n* = 3, *p* < 0.001; procollagen1A2: *n* = 3, *p* < 0.001; procollagen 3A1: *n* = 3, *p* < 0.001). Similarly, on day 21 after the completion of the model, the mRNA levels of Mfn2 in the rats in the POP group were significantly increased (*n* = 3, *p* = 0.012), and the expression of procollagen 1A1/1A2/3A1 was significantly decreased (procollagen1A1: *n* = 3, *p* < 0.001; procollagen1A2: *n* = 3, *p* < 0.001; procollagen3A1: *n* = 3, *p* < 0.001).

Western blot analysis showed that, on day 0, the expression of procollagen 1A1/1A2/3A1 in the uterosacral ligament of POP rats was significantly decreased (procollagen 1A1: *n* = 3, *p* < 0.001; procollagen1A2: *n* = 3, *p* = 0.002; procollagen 3A1: *n* = 3, *p* < 0.001), and Mfn2 expression was significantly increased (*n* = 3, *p* < 0.001) (Figure 1C). On day 21 after the model was established, all of the types of procollagen were significantly higher (*n* = 3, *p* < 0.001, all), and Mfn2 expression was significantly decreased (*n* = 3, *p* < 0.001).

### 3.2. Stem Cell Culture and Identification

The cultivation and identification of stem cells were crucial for subsequent animal experiments. In order to distinguish stem cells from normal cells (Figure 2A–C), CD29, CD90, and CD106 were detected, and the positive rates were 92.5%, 100.0%, and 91.1%, respectively. Secondly, the negative markers CD11b and CD45 were used to distinguish MSCs from hematopoietic stem cells (Figure 2D,E), and the positive rates were 0.080% and 0.030%, respectively. These results confirmed that the cells were rat BMSCs. In order to verify the differentiation potential of stem cells, we performed osteogenic and adipogenic differentiation assays. The cells in the group treated with adipogenic differentiation medium had multiple red round lipid droplets compared to the cells in the control group (Figure 2F), indicating that the cells have the ability to induce adipocyte differentiation. After osteoinduction and Alizarin Red S staining (Figure 2G), the stem cells showed red lumpy material, indicating that the stem cells had successfully differentiated into bone cells. Overall, the cells were identified as MSCs.

### 3.3. Downregulation of Mfn2

In order to regulate the expression of Mfn2, lentiviral vectors were constructed, and the virus titer was detected (Figure 3A). This experiment was divided into sh-Mfn2 (the RNAi transfection group) and sh-NC (the empty vector transfection inhibition group) groups. qRT-PCR was used to detect the mRNA levels of Mfn2 in the transfected stem cells, which were found to be reduced by 78% in the cells in the sh-Mfn2 group compared to the cells in the control groups (non-treated control (NC) and sh-NC groups) (Figure 3B, *p* < 0.05). There were no significant differences between the two groups. Western blotting was used to detect the protein level of Mfn2; compared to cells in the control groups (NC and sh-NC groups) (Figure 3C), Mfn2 expression was reduced by 76% in the cells in the sh-Mfn2 group (*p* < 0.05), with no significant difference between the two control groups. In conclusion, the stem cell transfection was successful.

### 3.4. Therapeutic Effect of Stem Cells on POP Rats

After the stem cells were injected into the uterosacral ligament of the rats in each group, the rats were sacrificed on days 0 (death immediately after injection), 7, 14, and 21. The uterosacral ligaments of the rats in each group were collected in order to detect the expression of Mfn2 and various types of procollagens by RNA and protein extraction. We found that the mRNA levels of Mfn2 and procollagen 1A1/1A2/3A1 were not significantly changed in either group 7 days after injection (Figure 4A,B). However, at day 14 and day 21 (Figure 4C,D), the mRNA level of the Mfn2 expression in the rats in the N4 group was significantly decreased compared to the rats in the N1 group (day 14, *n* = 3, *p* < 0.05; day 21: *n* = 3, *p* < 0.05). The expression of procollagen 1A1/1A2/3A1 mRNA in the rats in the N4 group was significantly increased (*n* = 3, *p* < 0.05, all). Similarly, compared to the rats in the P1 group (Figure 4C,D), the mRNA level of Mfn2 expression in the rats in the P4 group was significantly decreased (day 14, *n* = 3, *p* < 0.05; day 21: *n* = 3, *p* < 0.05). At the same time, the mRNA level of procollagen 1A1/1A2/3A1 in the rats in the P4 group was significantly increased (*n* = 3, *p* < 0.05, all).

Western blot analysis showed that, on days 0 and 7, compared to the rats in the NPOP group (N1, N2, N2, N4), the expression of Mfn2 in the rats in the POP group (P1, P2, P3, P4) was significantly increased (*n* = 3, *p* < 0.001, all), and the expression of procollagen was significantly decreased (*n* = 3, *p* < 0.001, all) (Figure 4E,F). On days 0 and day 7, there were no significant changes in the expression of Mfn2 and procollagen 1A1/1A2/3A1 in the N4 rats compared to the N1 rats. Similarly, there was no significant change in the expression of Mfn2 and procollagen 1A1/1A2/3A1 in the P4 rats compared to the P1 rats. However, on days 14 and day 21, the expression of Mfn2 in the N4 rats was significantly lower than that in the N1 rats (day 14: *n* = 3, *p* = 0.005; day 21: *n* = 3, *p* < 0.001), and the expression of procollagen 1A1/1A2/3A1 was significantly increased (day 14: procollagen1A1: *n* = 3, *p* < 0.001; procollagen1A2: *n* = 3, *p* < 0.001; procollagen3A1: *n* = 3, *p* < 0.001; day 21: procollagen1A1: *n* = 3, *p* < 0.001; procollagen1A2: *n* = 3, *p* < 0.001; procollagen3A1: *n* = 3, *p* = 0.001). Meanwhile, on days 14 and 21 (Figure 4E,F), the protein level of Mfn2 expression in the P4 rats was also significantly decreased compared to the P1 rats (day 14: *n* = 3, *p* = 0.005; day21: *n* = 3, *p* = 0.005), and the expression of procollagen 1A1/1A2/3A1 was significantly increased (day 14: procollagen1A1: *n* = 3, *p* = 0.031; procollagen1A2: *n* = 3, *p* = 0.001; procollagen3A1: *n* = 3, *p* < 0.001; day 21: procollagen1A1: *n* = 3, *p* < 0.001; procollagen1A2: *n* = 3, *p* < 0.001; procollagen3A1: *n* = 3, *p* = 0.001). These results indicated that the downregulation of Mfn2 by stem cells had a certain time-dependent therapeutic effect on POP rats.

## 4. Discussion

As the global elderly population increases, the incidence of POP is increasing. In order to explore the pathogenesis of POP, we conducted a series of in vivo and in vitro experiments [17,18,19,20]. We found that the expression of Mfn2 was increased and procollagens were decreased in the uterosacral ligaments of POP patients [17]. By culturing primary uterosacral ligament fibroblasts from POP and NPOP patients in vitro, we found that the expression of Mfn2 was significantly increased and the expression of procollagen1A1/1A2/3A1 was significantly decreased in POP patients [18]. At the same time, we also confirmed that the downregulation of Mfn2 expression increased the proliferation of fibroblasts and the expression of procollagen1A1/1A2/3A1. Moreover, this process likely involves the Ras/Raf/ERK pathway [19]. Therefore, we speculated that the downregulation of Mfn2 expression could treat POP. In the present study, this hypothesis was verified using animal experiments. This study successfully completed the culture and identification of rat BMSCs, and the construction of POP rat models. We injected stem cells that inhibited Mfn2 expression into the uterosacral ligaments of rats, and then collected the uterosacral ligaments in order to detect the expression of Mfn2 and procollagen 1A1/1A2/3A1. We found that stem cells could inhibit the mRNA and protein expression of Mfn2 in the uterosacral ligament of POP rats, and that they promoted the expression of procollagen 1A1/1A2/3A1.

This study is novel in two ways; first, previous studies [18,19] on Mfn2 mostly focused on cardiovascular and cerebrovascular diseases, and diabetes, etc., and there is little research on Mfn2 and pelvic floor diseases [14]. POP is a common disease in middle-aged and elderly individuals. Mfn2 is related to aging, and it is generally recognized that Mfn2 is related to the damage and aging of the pelvic floor’s supporting structure. Therefore, we proposed an innovative hypothesis: Mfn2 is involved in the metabolism and functional regulation of pelvic floor fibroblasts. The abnormal expression of Mfn2 can inhibit cell proliferation and promote apoptosis by mediating abnormal mitochondrial fusion, or by regulating intracellular signaling, resulting in a decrease in procollagen synthesis in fibroblasts and a decrease in collagen secretion in pelvic floor supporting tissue, which eventually lead to POP in this work. This hypothesis was preliminarily verified in our previous studies [17,18]. Second, the existing treatment methods for POP have many problems [7,8]. Therefore, there is an urgent need to explore safer and more effective treatment methods. Stem cells are safer than traditional gene regulation methods. In a review of recent successful stem cell clinical trials, researchers [22] found that HIV-derived, self-inactivated lentiviral vectors are less toxic, make gene transfection more effective, and allow for stable expression, which are desirable in the field of gene therapy. Moreover, stem cells have the potential for self-renewal and directional differentiation. In this field, some researchers [14,25,26] have conducted corresponding studies on induced pluripotent stem cells, BMSCs, and endometrial stromal cells, which showed certain therapeutic effects. However, these studies focused mostly on operations combined with meshes. Few studies [18] have combined the advantages of stem cell gene regulation to regulate the expression of specific genes involved in POP. At the same time, the effect of the downregulation of Mfn2 expression on POP has not been discussed. In this study, we provided novel evidence for the prevention and treatment of POP by the regulation of the expression of Mfn2 at the stem cell level.

POP rat models were established by ovariectomy and stretch dilation. We found that compared to the rats in the NPOP group, the expression of Mfn2 in the uterosacral ligament of the rats in the POP group was significantly increased, and the expression of procollagen1A1/1A2/3A1 was significantly decreased. This indicated that the rat model was successfully constructed, which was consistent with the characteristics of the uterosacral ligament tissues and primary fibroblasts of POP patients [17,18,19,20]; this further verified that Mfn2 plays an important role in the occurrence and development of POP. The stem cells functioned in a time-dependent manner, and we found that the mRNA and protein levels of Mfn2 and procollagen1A1/1A2/3A1 in the N4 and P4 rats were not significantly different from those in the N1 and P1 rats on days 0 and day 7. When we extended the observation time, the expressions of Mfn2 and procollagen 1A1/1A2/3A1 in the rats in the N4 and P4 groups were significantly changed compared to the rats in the N1 and P1 groups on days 14 and day 21. These results indicated that stem cells with the ability to downregulate Mfn2 expression could increase procollagen 1A1/1A2/3A1 expression in the uterosacral ligament of POP rats, which was helpful in increasing the pelvic floor support, thus providing a therapeutic benefit.

A study limitation is that we only observed the curative effect at 21 days post-injection, and not over a longer timescale, such as three months or even half a year, which warrants further investigation. This also depends on clinical transformation experiments, and the clinical transformation of basic research remains a challenge. In future studies, data related to the distribution and function of stem cells and the mechanism of action in host tissues are key to the realization of their clinical value [23], which is also a bottleneck for large-scale clinical application. In addition, this study is still a preliminary exploration. In the future, we intend to further search for RNAs and proteins related to Mfn2 through transcriptomics and proteomics, so as to explore their interaction and explain the pathogenesis of POP.

In conclusion, the downregulation of Mfn2 in the uterosacral ligament of POP rats can promote the expression of collagens, improve the support strength of pelvic floor tissue, and alleviate POP to some extent. This approach demonstrates a novel therapeutic approach to the prevention and treatment of POP.

## Figures and Tables

**Figure 1 genes-13-00829-f001:**
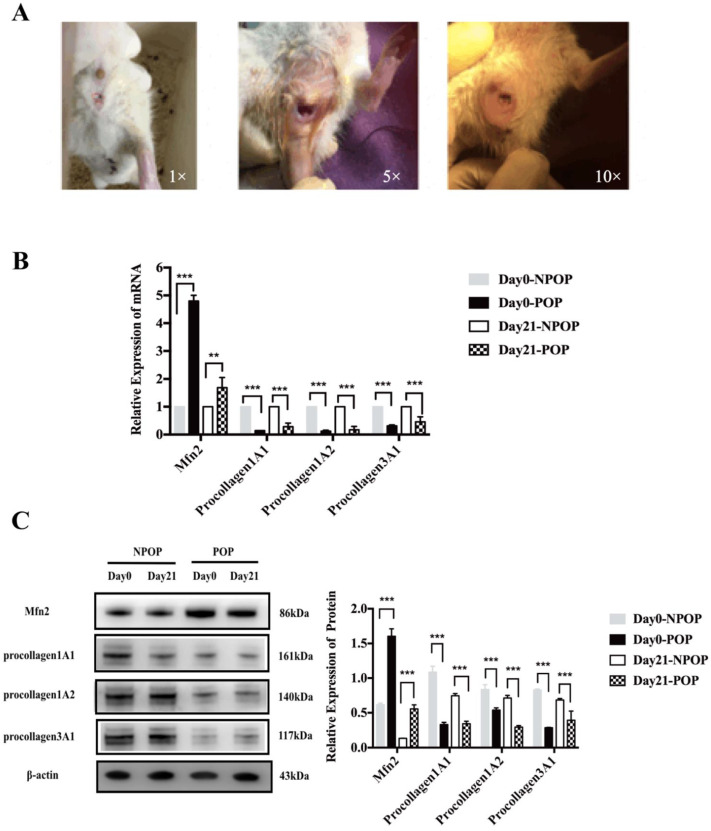
Establishment of the POP rat model. (**A**) From left to right, after one, five, and 10 stretching cycles. (**B**) After the model was established, the uterosacral ligaments in rats in the POP and NPOP groups were collected on days 0 and 21 for qRT-PCR detection. (**C**) After the model was established, the uterosacral ligaments in the rats in the POP and NPOP groups were collected on days 0 and 21 for Western blot detection. ** *p* < 0.01, *** *p* < 0.001.

**Figure 2 genes-13-00829-f002:**
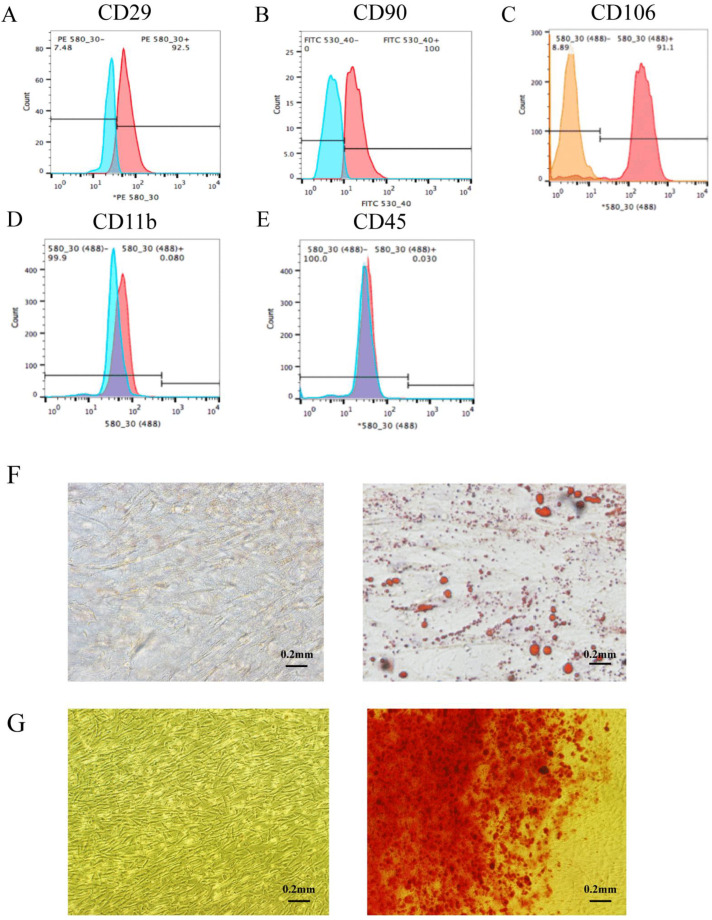
Culture and identification of rat BMSCs. (**A**–**C**) The positive markers were used to differentiate the stem cells from other cells. The positive rates of CD29, CD90 and CD106 were 92.5%, 100% and 91.1%, respectively. Red line represents the antibody to be tested, and blue or orange line represents the homotypic control. (**D**,**E**) The negative markers CD11b and CD45 were used to distinguish mesenchymal stem cells from hematopoietic stem cells. The positive rates of CD11b and CD45 were 0.080% and 0.030%, respectively. Red line represents the antibody to be tested, and blue line represents the homotypic control. (**F**) Adipogenic induction and Oil Red O staining; (**G**) Osteogenic induction and Alizarin Red S staining.

**Figure 3 genes-13-00829-f003:**
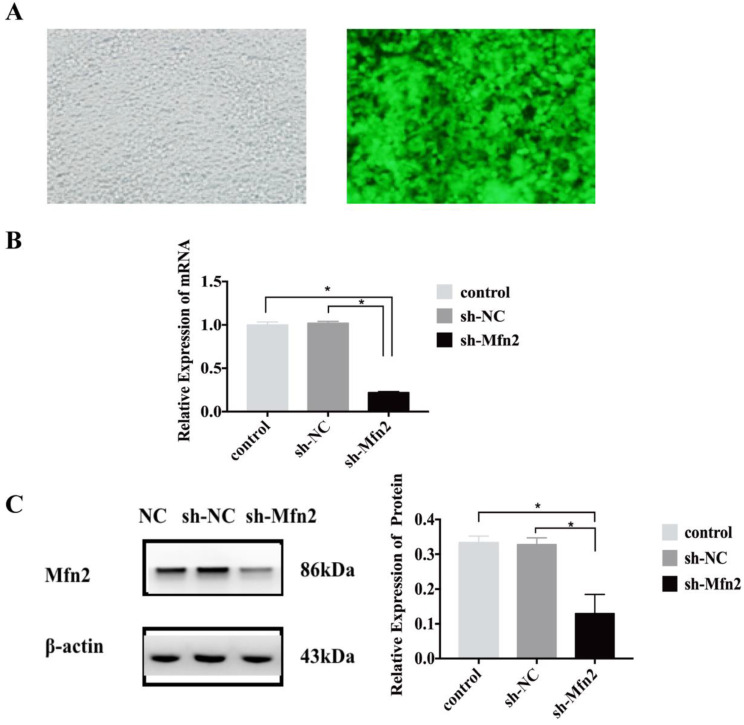
Inhibition of Mfn2 expression. (**A**) Viral titer detection after the inhibition of Mfn2. Left: bright field, 200×; Right: green fluorescence field, 200×; detected cells: 293T cells. (**B**) After the sh-Mfn2 and sh-NC lentiviruses were transfected into stem cells, the mRNA level of Mfn2 was detected by qRT-PCR. (**C**) After the sh-Mfn2 and sh-NC lentiviruses were transfected into stem cells, the protein expression level of Mfn2 was detected by western blot. * *p* < 0.05. All of the experiments were repeated three times.

**Figure 4 genes-13-00829-f004:**
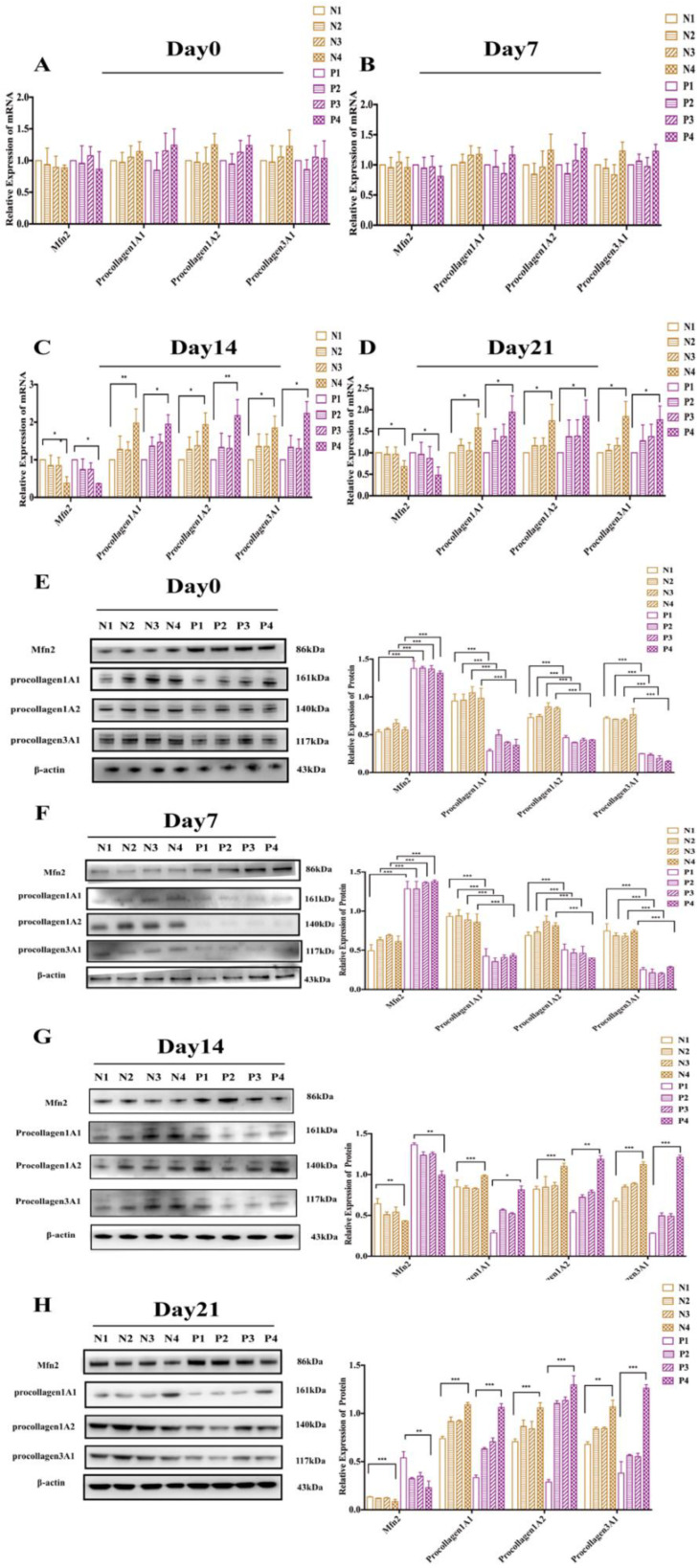
Therapeutic effect of stem cells on POP rats. (**A**) On day 0 of the injection treatment, the uterosacral ligaments of the rats in the eight groups were collected for mRNA detection; (**B**) mRNA analysis on day 7 post-stem cell injection; (**C**) mRNA analysis on day 14 post-stem cell injection; (**D**) mRNA analysis on day 21 post-stem cell injection; (**E**–**H**) Western blot analysis on days 0, 7, 14, and 21 post-stem cell injection. * *p* < 0.05, ** *p* < 0.01, *** *p* < 0.001. All of the experiments were repeated three times.

## Data Availability

The data that support this study are available from the corresponding author upon reasonable request.

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
