# Peer review of "Treatment of Pelvic Organ Prolapse by the Downregulation of the Expression of Mitofusin 2 in Uterosacral Ligament Tissue via Mesenchymal Stem Cells"

_genes, 2022, doi:10.3390/genes13050829_

Round 1

Reviewer 1 Report

The authors presented original experimental study on treatment of pelvic organ prolapse using stem cells. Role of mitofusin 2 (Mfn2) expression is shown, following the discussion of this possible new gene target in the treatment. This new original study supported by unique experimental data should be published in the journal.

However I have series of the remarks to be taken into account before the publication.

First, the formatting of the manuscript should be fixed to follow the journal standards.

Section 2.4 is formatted as long title.

The reference list is not formatted (extra numbering like ‘1.  [1].’ In the references, extra signs ‘,;’ in all the authors names)  ). Formal paragraphs like

‘Supplementary Materials: The following are available online at www.mdpi.com/xxx/s1, Figure S1: title’ - just not filled.

The citations in the text are in bulk - like (1-8), (9-14) - too many in-text references together.

The ‘p-values = 0.000’ wording throughout the text should be corrected. Either indicate exact p-value, or write ‘p<0.001’ not using ‘0.000’ - is it not correct, not in the publication standards.

Major remark - while the animal model is described in details the selection of Mfn2 for the study is not well explained. Why not measure expression of several genes (or transcriptome) in the experiments? Only some  procollagen genes are mentioned..

Please add some corresponding discussion to the text.

Minor remarks:

First phrase in the Abstract - line 9-10:

‘As an aging-related disease, the relationship between pelvic organ prolapse  (POP) and the aging-related protein, mitofusin 2 (Mfn2), has rarely been studied.’ - separate in two sentences. Logically these statements are different. And wording ‘age-related’ is repeated twice. Need to rewrite

Line 14: ‘four respective groups’ - here are 8 groups, need too rephrase.

Or write like ‘in 4 groups for negative controls (N1-N4), and 4 groups for prolapse (P1-P4), respectively’. Change phrase about the group number in the abstract and similar phrase in the main text.

Indicate total number of animals used (I see n=3 in the text, does it means total number is 12 for negative controls and 12 for prolapse treatment?)

Line 23 ‘(P = 0.000, all).’ - remove zeros, or write ‘<0.001’

Line 38 - keywords - add ‘animal model’ or ‘rat model’ instead of just ‘Rat’ keyword

Line 41: citations like (1-8) is not acceptable. It is not good to cite more than 3 references together in the text without details. Either cite less, remove extra references, or add details about the disease.

Line 43: ‘is limited (9-14).’ - again too many references together. Add details. New phrases about current studies to provide details. Or cite less references together.

Line 50: ‘(17-20)’ - again bulk citations. This set of references is repeated in the text. Describe previous findings in details. Otherwise it looks like self-citations.

May add the reference here to the phrase in line 52: ‘Our previous studies also confirmed that... procollagen...’

Lines 64 and 66: ‘(14,21-25).’  and ‘(14,21-31)’ - bulk citations again. Please cite only relevant papers each time, like ‘stem cells (14,25,26)’ in line 69.

Section ‘2.1. Construction of the POP animal model’ should have some reference to the protocol, or previous publications. Please add details, indicate total number of animal used, and counted in the experimental groups.

Need to prove the castration procedure is really mimic the hormone levels by citing some previous publications.

Line 89: ‘operation line No. 0’ - please comment what is No.0?

Line 123: ‘(CD29’ and below ‘(CD11b’ - need remove parentheses or add closing parenthesis ‘)’ - please check the formatting.

Line 126: ‘FlowJo software’ - need reference for the software.

Section 2.4 - change the font, add proper section title (like ‘Lipid forming capacity’ to this section)

Line 152: ‘experimental lentivirus was designed...’ - is it unique lentivirus designed only for this work, or it was used before? Need add a reference then.

Line 164: ‘into four groups: N1:’ - add word ‘correspondingly’, indicate that there are 4 groups for control and treatment (POP), 8 groups in total.  Have  you used technical replicates for each group?

Line 171: ‘2.7. Real-time quantitative PCR’ - it is interesting to see gene expression for other genes. Why only Mfn2 expression was measured? May comment on other available expression data, if any..

Line 179: ‘2 – ΔΔCT’ - check superscript font formatting

I’d recommend write all the primers in a small table. It is hard to read just in the text. But, may keep the primers description as is. Just fix spaces in writing ‘ -3’’ (remove extra spaces)

Line 189: ‘with RIPA’ - give the abbreviation in full.

Line 190: ‘the BCA method’ - give the abbreviation of the method in full, add a reference.

Line 201: ‘ImageJ software’ - add a reference (or web-link) about the software, indicate version.

Line 204: ‘SPSS23 software’ - add a reference.

Line 206: ‘K-W test’ - write ‘K-W’ in full.

Section ‘3.1. Rat model of POP’ - write total number of rats used.

Remove zeros in ‘P=0.000’ wording, write exact values, or ‘<0.001’

Figure 1 - panel A. Add to the figure legend (line 230) what one can see in the photos  ‘1X’, ‘5X’, ‘10X’ ? Assume after prolapse treatment could be some visible changes ?

Figure 2. Add panel letter ‘D’, ‘E’, ‘H’ to the figure graphics. Currently the figure legend does not correspond to the images.

Line 274: ‘day  0 (death immediately after injection’ - may comment on it. Does it mean no time (even half an hour after injection)? Physiologically should be some time after injection to measure effects.

Line 294: ‘P=0.000’ - remove extra zeros.

Line 309: ‘E-F:’ - only panels E and F are described in the figure legend. No F and H letters. I think it is a typo...

Line 315: ‘(17-20)’ - bulk citation again. Try to avoid it. The citations by each reference separately in the same paragraph - (1) and (18) is quite detailed. It could cited by the same manner from the Introduction.

Line 330: ‘previous studies’ - here need add references.

Line 331: ‘little research on’ - here should be references on the existing research too..

Line 334: ‘Therefore, we proposed an innovative hypothesis...’ - add a reference to previous work or add wording ‘for the first time’, ‘in this work’, ‘here’ to show that it is novel hypothesis only suggested in current work.

Line 340: ‘study  (17-20)’ - change to ‘studies’ - plural form.

Line 341: ‘the existing ... methods ... have many problems’ - add reference on existing methods and the problems.

Section 394: ‘Data Availability Statement: The data... The datasets...’ - the phrases are repeat the same. Please fix the statement in this section by single phrase.

The references - please remove extra numbering ‘1.  [1].’ ‘2.  [2].’ And so on for all the reference formatting.

Remove extra commas and semicolons ,; -

For example “1.  [1]. Khan AA,; Eilber KS,; Clemens JQ,; Wu N,; Pashos CL,; Anger JT”

Should be “1. Khan AA, Eilber KS, Clemens JQ, Wu N,;Pashos CL, Anger JT”

Check in the references down.

Please check also Chinese names spelling - should not be mixture of capital letters in the same name. For example ‘XiaoQing Wang’ could be written as ‘Wang XQ or ‘Wang Xiaoqing’ or ‘Wang Xiao-Qing’ just to follow the standard of English translation. Though may write as is it was in previous publications. I see, for example, in Frontiers in genetics ‘Xiao-Qing Wang’ writing.

Author Response

First, the formatting of the manuscript should be fixed to follow the journal standards.

Section 2.4 is formatted as long title.

Thanks a lot for your constructive suggestion and kind reminding, which is highly appreciated. We have added an appropriate title here(line134).

The reference list is not formatted (extra numbering like ‘1.  [1].’ In the references, extra signs ‘,;’ in all the authors names)  ). Formal paragraphs like

Thank you for your comments. We have taken this good advice and We have made corresponding modifications (line415-470).

‘Supplementary Materials: The following are available online at www.mdpi.com/xxx/s1, Figure S1: title’ - just not filled.

Thanks a lot for your constructive suggestion and kind reminding. We have added the title.

The citations in the text are in bulk - like (1-8), (9-14) - too many in-text references together.

We appreciate your constructive comments. We added relevant details for better reference(line 41-46).

The ‘p-values = 0.000’ wording throughout the text should be corrected. Either indicate exact p-value, or write ‘p<0.001’ not using ‘0.000’ - is it not correct, not in the publication standards.

Thank you for your comments. We have taken this good advice accordingly (line all).

Major remark - while the animal model is described in details the selection of Mfn2 for the study is not well explained. Why not measure expression of several genes (or transcriptome) in the experiments? Only some procollagen genes are mentioned. Please add some corresponding discussion to the text.

Thanks a lot for your constructive suggestion and kind reminding, which is highly appreciated. In fact, before this study, we conducted a large number of in vivo and in vitro tests to confirm the relationship between Mfn2, procollagens and POP. We think that this part of experiment may not be optimal, but should be sufficient to draw a conclusion that downregulation of Mfn2 expression increased the proliferation of fibroblasts and the expression of procollagen1A1/1A2/3A1 in POP patients. Therefore, we directly verified these proteins we are most concerned about in animal experiments. However, this study is still a preliminary exploration. In the future, we intend to further search for RNAs and proteins related to Mfn2 through transcriptomics and proteomics, so as to explore their interaction and explain the pathogenesis of POP. We also supplemented the discussion accordingly(line 388-391).

Minor remarks:

First phrase in the Abstract - line 9-10:

‘As an aging-related disease, the relationship between pelvic organ prolapse (POP) and the aging-related protein, mitofusin 2 (Mfn2), has rarely been studied.’ - separate in two sentences. Logically these statements are different. And wording ‘age-related’ is repeated twice. Need to rewrite

We have taken this good advice and amended it accordingly (line 10).

Line 14: ‘four respective groups’ - here are 8 groups, need too rephrase.

Or write like ‘in 4 groups for negative controls (N1-N4), and 4 groups for prolapse (P1-P4), respectively’. Change phrase about the group number in the abstract and similar phrase in the main text.

We appreciate your advice and are especially willing to add the sentence to the article (line 14-17).

Indicate total number of animals used (I see n=3 in the text, does it means total number is 12 for negative controls and 12 for prolapse treatment?)

Thank you very much for your question. The total number of POP rats was added(line220). There are four time points (D0, D7, D14, D21). There were 24 rats at each time point. Among them, group N1 = 3, N2 = 3, N3 = 3, N4 = 3, P1 = 3, P2 = 3, P3 = 3 and P4 = 3.

Line 23 ‘(P = 0.000, all).’ - remove zeros, or write ‘<0.001’c

We truly appreciate the reviewer’s professional question and we have revised it as reviewer suggested (line23).

Line 38 - keywords - add ‘animal model’ or ‘rat model’ instead of just ‘Rat’ keyword

 We truly appreciate the reviewer’s kindly suggestion and we have revised it (line37).

Line 41: citations like (1-8) is not acceptable. It is not good to cite more than 3 references together in the text without details. Either cite less, remove extra references, or add details about the disease. Line 43: ‘is limited (9-14).’ - again too many references together. Add details. New phrases about current studies to provide details. Or cite less references together.

We appreciate your constructive comments. We added relevant details for better reference(line 41-46).

Line 50: ‘(17-20)’ - again bulk citations. This set of references is repeated in the text. Describe previous findings in details. Otherwise it looks like self-citations.

We appreciate your constructive comments. We have delete citations accordingly(line 53-59).

May add the reference here to the phrase in line 52: ‘Our previous studies also confirmed that... procollagen...’

We truly appreciate the reviewer’s kindly suggestion and we have revised it (line55).

Lines 64 and 66: ‘(14,21-25).’  and ‘(14,21-31)’ - bulk citations again. Please cite only relevant papers each time, like ‘stem cells (14,25,26)’ in line 69.

We appreciate your constructive comments. We have delete citations accordingly(line 67-70).

Section ‘2.1. Construction of the POP animal model’ should have some reference to the protocol, or previous publications. Please add details, indicate total number of animal used, and counted in the experimental groups.

Thank you for your constructive suggestion. We appreciate your advice and are especially willing to add citation to the article(83-84, 92).

Need to prove the castration procedure is really mimic the hormone levels by citing some previous publications.

Thank you for your constructive suggestion. We appreciate your advice and are especially willing to add citation to the article(86).

Line 89: ‘operation line No. 0’ - please comment what is No.0?

We truly appreciate the reviewer’s professional question and we have revised it as reviewer suggested (line 93).

Line 123: ‘(CD29’ and below ‘(CD11b’ - need remove parentheses or add closing parenthesis ‘)’ - please check the formatting.

We truly appreciate the reviewer’s professional question and we have remove parentheses as reviewer suggested (line130).

Line 126: ‘FlowJo software’ - need reference for the software.

Thank you for your constructive suggestion. We have revised it and hope that you will now be satisfied(line 131).

Section 2.4 - change the font, add proper section title (like ‘Lipid forming capacity’ to this section)

Thanks a lot for your constructive suggestion and kind reminding, which is highly appreciated. We have changed the font and added an appropriate title here. (line134-146)

Line 152: ‘experimental lentivirus was designed...’ - is it unique lentivirus designed only for this work, or it was used before? Need add a reference then.

We truly appreciate the reviewer’s professional question. The unique lentivirus was designed only for this work.

Line 164: ‘into four groups: N1:’ - add word ‘correspondingly’, indicate that there are 4 groups for control and treatment (POP), 8 groups in total.  Have you used technical replicates for each group?

The word ‘correspondingly’ was added(line171). We used technical replicates for each group.

Line 171: ‘2.7. Real-time quantitative PCR’ - it is interesting to see gene expression for other genes. Why only Mfn2 expression was measured? May comment on other available expression data, if any.

Thanks a lot for your constructive suggestion and kind reminding, which is highly appreciated. In fact, before this study, we conducted a large number of in vivo and in vitro tests to confirm the relationship between Mfn2, procollagen and POP. We think that this part of experiment may not be optimal, but should be sufficient to draw a conclusion that downregulation of Mfn2 expression increased the proliferation of fibroblasts and the expression of procollagen1A1/1A2/3A1 in POP patients. Therefore, we directly verified these proteins we are most concerned about in animal experiments. However, this study is still a preliminary exploration. In the future, we intend to further search for RNAs and proteins related to Mfn2 through transcriptomics and proteomics, so as to explore their interaction and explain the pathogenesis of POP. We also supplemented the discussion accordingly (line 389-391).

Line 179: ‘2 – ΔΔCT’ - check superscript font formatting

I’d recommend write all the primers in a small table. It is hard to read just in the text. But, may keep the primers description as is. Just fix spaces in writing ‘ -3’’ (remove extra spaces)

We have revised the superscript font formatting and extra spaces were removed(line 188-195).

Line 189: ‘with RIPA’ - give the abbreviation in full.

Thank you for your constructive suggestion. We have revised it and hope that you will now be satisfied (line 198-199).

Line 190: ‘the BCA method’ - give the abbreviation of the method in full, add a reference.

Thank you for your constructive suggestion. We have revised it (line 199).

Line 201: ‘ImageJ software’ - add a reference (or web-link) about the software, indicate version.

Thank you for your constructive suggestion. We have revised it (line 210).

Line 204: ‘SPSS23 software’ - add a reference.

 Thank you for your constructive suggestion. We have revised it (line 213).

Line 206: ‘K-W test’ - write ‘K-W’ in full.

Thank you for your constructive suggestion. We have revised it and hope that you will now be satisfied(line 215-216).

Section ‘3.1. Rat model of POP’ - write total number of rats used.

The total number of rats was added(line220).

Remove zeros in ‘P=0.000’ wording, write exact values, or ‘<0.001’

 We truly appreciate the reviewer’s professional question and we have revised it as reviewer suggested (line all).

Figure 1 - panel A. Add to the figure legend (line 230) what one can see in the photos  ‘1X’, ‘5X’, ‘10X’ ? Assume after prolapse treatment could be some visible changes ?

Thanks a lot for your constructive suggestion and kind reminding, we added relevant descriptions(221-224).

Figure 2. Add panel letter ‘D’, ‘E’, ‘H’ to the figure graphics. Currently the figure legend does not correspond to the images.

Thanks a lot for your constructive suggestion and kind reminding, we added relevant descriptions(line 261-264).

Line 274: ‘day0 (death immediately after injection’ - may comment on it. Does it mean no time (even half an hour after injection)? Physiologically should be some time after injection to measure effects.

Thanks a lot for your constructive suggestion and kind reminding. In fact, in order to analyze the relationship between stem cell function and time, we set up this group. This set of data plays a control role to some extent.

Line 294: ‘P=0.000’ - remove extra zeros.

Thank you for your comments. We have taken this good advice accordingly(line 299).

Line 309: ‘E-F:’ - only panels E and F are described in the figure legend. No F and H letters. I think it is a typo...

 We truly appreciate the reviewer’s professional question and we have revised it (line 320-322).

Line 315: ‘(17-20)’ - bulk citation again. Try to avoid it. The citations by each reference separately in the same paragraph - (1) and (18) is quite detailed. It could cited by the same manner from the Introduction.

We appreciate your constructive comments. We have delete citations accordingly(line 340).

Line 330: ‘previous studies’ - here need add references.

We appreciate your constructive comments. We have added citations accordingly(line 352).

Line 331: ‘little research on’ - here should be references on the existing research too.

We appreciate your constructive comments. We have rephrased the sentence of the discussion accordingly(line343). 

Line 340: ‘study (17-20)’- change to ‘studies’ - plural form.

Thank you for your constructive suggestion. We truly appreciate the reviewer’s professional question. We have delete citations accordingly(line 341).

Line 341: ‘the existing ... methods ... have many problems’ - add reference on existing methods and the problems.

 We have added citations accordingly (line352).

Section 394: ‘Data Availability Statement: The data... The datasets...’ - the phrases are repeat the same. Please fix the statement in this section by single phrase.

We appreciate your constructive comments. We have rephrased the sentences accordingly(line408-409).

The references - please remove extra numbering ‘1.  [1].’ ‘2.  [2].’ And so on for all the reference formatting.

Remove extra commas and semicolons ,; -

For example “1.  [1]. Khan AA,; Eilber KS,; Clemens JQ,; Wu N,; Pashos CL,; Anger JT”

Should be “1. Khan AA, Eilber KS, Clemens JQ, Wu N,;Pashos CL, Anger JT”

Check in the references down.

We appreciate your constructive comments. Extra commas and semicolons were removed (line 415-470).

Please check also Chinese names spelling - should not be mixture of capital letters in the same name. For example ‘XiaoQing Wang’ could be written as ‘Wang XQ or ‘Wang Xiaoqing’ or ‘Wang Xiao-Qing’ just to follow the standard of English translation. Though may write as is it was in previous publications. I see, for example, in Frontiers in genetics ‘Xiao-Qing Wang’ writing.

Thanks a lot for your constructive suggestion and kind reminding. Chinese names spelling were checked (line 5-6).

Reviewer 2 Report

The authors present interesting research involving the application of mesenchymal stem cells in the treatment of pelvic organ prolapse. This disorder occurs very often in older women; thus, the idea of new and alternative protocols is of high importance. However, there are some issues and remarks to be clarified: 
(1) the methodology description should be improved, i.e., add details of the scientific equipment used in the study (microscope,qRT-PCR). 
(2) describe in more detail the conditions of harvesting stem cells (CO2 or O2?), humidity, what kind of bottles, add the company of cell culture plastics used. Were cells tested for mycoplasm?
(3) how many animals were used per one group?
(4) line 129-139 - italic? Additional section?
(5) line 159 - there is stated ". GFP expression in cells was observed...", in which step GFP was incorporated in lentivirus? It is not clear from where GFP comes. 
(6) line 171 - how was RNA isolated, and was the quality measured?
(7) there is a lack of SD in FIG.1B., and blots images are of insufficient resolution
(8) fic. 3 B and C, poor resolution of legends and axis description 
(9) fig 4 - graphs are not readable; consider different representations, bigger fonts. 

Author Response

(1) the methodology description should be improved, i.e., add details of the scientific equipment used in the study (microscope, qRT-PCR). 

Thank you for your comments. We have taken this good advice and supplemented the details accordingly (line 113-114,182).

(2) describe in more detail the conditions of harvesting stem cells (CO2 or O2?), humidity, what kind of bottles, add the company of cell culture plastics used. Were cells tested for mycoplasm?

We have taken this good advice and add them accordingly(line 108-109). The cells were tested for mycoplasma.

(3) how many animals were used per one group?

 Thanks a lot for your constructive suggestion and kind reminding. The total number of POP rats was added(line220). There are four time points (D0, D7, D14, D21). There were 24 rats at each time point. Among them, group  N1 = 3, N2 = 3, N3 = 3, N4 = 3, P1 = 3, P2 = 3, P3 = 3 and P4 = 3.

(4) line 129-139 - italic? Additional section?

Thank you for your constructive suggestion. We truly appreciate the reviewer’s professional question. This part should not be italicized, and we changed the font (line 135-146).

(5) line 159 - there is stated ". GFP expression in cells was observed...", in which step GFP was incorporated in lentivirus? It is not clear from where GFP comes. 

We appreciate your constructive comments. We have add the information accordingly (line 159).

(6) line 171 - how was RNA isolated, and was the quality measured?

Thanks a lot for your constructive suggestion and kind reminding, which is highly appreciated. We have taken this good advice and add the information accordingly (line 180-183).

(7) there is a lack of SD in FIG.1B., and blots images are of insufficient resolution

We truly appreciate the reviewer’s professional question. We have taken this good advice and amended Fig.1 accordingly. Although the blots images are of insufficient resolution, they are enough to illustrate the problem and  relatively convincing figures we can provide.

(8) fic. 3 B and C, poor resolution of legends and axis description

We agree with these valuable suggestions, which are very helpful to improve the quality of our articles. We have taken this good advice and amended Fig.3 accordingly.

(9) fig 4 - graphs are not readable; consider different representations, bigger fonts.

We have taken this good advice and amended Fig.4 accordingly.
